# Comparative Analysis of Gut Microbiomes in Laboratory Chinchillas, Ferrets, and Marmots: Implications for Pathogen Infection Research

**DOI:** 10.3390/microorganisms12040646

**Published:** 2024-03-24

**Authors:** Jindan Guo, Weixiong Shi, Xue Li, Bochao Yang, Chuan Qin, Lei Su

**Affiliations:** NHC Key Laboratory of Human Disease Comparative Medicine, Beijing Engineering Research Center for Experimental Animal Models of Human Critical Diseases, International Center for Technology and Innovation of Animal Model, Institute of Laboratory Animal Sciences, Chinese Academy of Medical Sciences (CAMS) & Comparative Medicine Center, Peking Union Medical College (PUMC), Beijing 100021, China; guojd@mail.bnu.edu.cn (J.G.); shiweixiong@cnilas.org (W.S.); lixue@cnilas.org (X.L.); yebeek@126.com (B.Y.); qinchuan@pumc.edu.cn (C.Q.)

**Keywords:** high-throughput sequencing, feeding habitat, gut mycobiome, interaction networks, diversity, chinchillas, ferrets, marmots

## Abstract

Gut microbes play a vital role in the health and disease of animals, especially in relation to pathogen infections. Chinchillas, ferrets, and marmots are commonly used as important laboratory animals for infectious disease research. Here, we studied the bacterial and fungal microbiota and discovered that chinchillas had higher alpha diversity and a higher abundance of bacteria compared to marmots and ferrets by using the metabarcoding of 16S rRNA genes and ITS2, coupled with co-occurrence network analysis. The dominant microbes varied significantly among the three animal species, particularly in the gut mycobiota. In the ferrets, the feces were dominated by yeast such as *Rhodotorula* and *Kurtzmaniella*, while in the *chinchillas*, we found *Teunomyces* and *Penicillium* dominating, and *Acaulium*, *Piromyces*, and *Kernia* in the marmots. Nevertheless, the dominant bacterial genera shared some similarities, such as *Clostridium* and *Pseudomonas* across the three animal species. However, there were significant differences observed, such as *Vagococcus* and *Ignatzschineria* in the ferrets, *Acinetobacter* and *Bacteroides* in the chinchillas, and *Bacteroides* and *Cellvibrio* in the marmots. Additionally, our differential analysis revealed significant differences in classification levels among the three different animal species, as well as variations in feeding habitats that resulted in distinct contributions from the host microbiome. Therefore, our data are valuable for monitoring and evaluating the impacts of the microbiome, as well as considering potential applications.

## 1. Introduction

With the development of life science, laboratory animals are playing vital roles in scientific research, teaching, production, verification, safety evaluation, and achievement assessment. However, research on these animals is still in its early stages. To better utilize and develop their value, three special laboratory animals—marmot, chinchilla, and ferret—were chosen for their large potential in the study of human infectious diseases and viral therapeutic development as essential animal models [1,2,3,4]. For example, the chinchilla model of otitis media could be used to study extracellular DNA that protects resident bacteria from antibiotics and host immune effectors [5]. The marmot infected by the hepatitis virus (WHV) represents an informative animal model for studying HBV infection and HCC [6]. Ferrets are widely employed to study the pathogenicity, transmissibility, and tropism of influenza viruses [7]. These animal models play a crucial role in studying human disease infections.

The gut microbiota, which can affect the physiological and pathological states of the host by modulating the host’s metabolism and immune system [8], has gained more attention [9]. In-depth research on the gut microbiota, which involves a large and complex microbial community, is an important part of infectious disease control and has numerous applications. Gut microbiota diversity is often associated with factors different from body mass, including diet [10], digestive tract structure [11], state of health (immune system, pregnancy, obesity) [12], or genetic background [13]. Although there are related gut microbiota studies in marmots [14], chinchillas [15], and ferrets [16], information on the gut microbiota’s inter- and intra-specific diversity is still lacking [17]. Mammals have existed on Earth for millions of years, and mammals with different feeding habits have different digestive tract structures due to extensive evolutionary radiation. The three special laboratory animals mentioned before have obviously different features, such as different digestive tract structures, which could contribute to the microbial community structure and composition. In general, mammals have a longer small intestine than large intestine, except for chinchillas. Compared with marmots, chinchillas have a larger volume of the cecum and a greater relative length of the hindgut. Although marmots are herbivorous animals, they sometimes eat small animals such as insects and exhibit omnivorous behavior under breeding conditions. Ferrets, as carnivores, are small animals with an extremely fast metabolism and require a high-protein diet with very low carbs and sugars. Their stomachs are quite large, and their intestines are intricate, even though they are very short. Unlike humans and some other animals, ferrets do not have a cecum or ileocolic.

Bacteria, as the most abundant microbes in the gut of mammals, are more significant than other members of the gut microbiota and have thus been the focus of most research [18]. In addition to bacteria, other microbes like fungi [19], bacteriophages [20], and archaea [21] also play an irreplaceable role and form the gut microbiota together with bacteria. Currently, most research on the microbiome focuses on the bacterial component [22], leading to a lack of information about other microbial communities, especially the fungal component (mycobiota), and how they can influence host health [23]. In fact, fungi, as key microbes closely related to feeding habitat, with most foods positively or negatively associated, can affect animal growth, development, systemic evolution, nutrient acquisition, cellulose degradation, and fermentation [24].

In this study, we researched both the gut bacterial and fungal communities in the three animal species using high-throughput sequencing. We explored the composition, diversity, and microbial roles of the gut microbiota shaped by the host species and feeding habitat. Moreover, we constructed networks of intra- and inter-dominant bacterial and fungal genera. These results provide a better understanding of environment–diet–microbe–host interactions. Furthermore, the change in host health status reflects the complex balance relationship between the host, symbiotic flora, and pathogen. Usually, gut microbes can directly interact with pathogens and directly inhibit or promote the process of virus invasion or replication. At present, the impact of the host microbiome on infection outcomes has not been explored in animal models, partly due to the lack of a comprehensive understanding of microbial communities across different laboratory animal species. Research on the diversity of gut bacteria and fungi in these experimental animals can lay the foundation for their application in future medical research.

## 2. Materials and Methods

### 2.1. Animals and Sampling Scheme

No human subjects participated in this study. Fecal samples were collected from fifteen healthy animals of three species (*Chinchilla lanigera*, *Marmota bobak*, and *Mustela putorius furo*), which were raised in the Institute of laboratory animal sciences, Chinese academy of medical sciences. Information on fecal samples, including age, sex, and sampling date, is provided in Appendix A. Prior to the fecal sampling, all animals were fed a forage-based diet for 1 month (Appendix A). Water was given once daily. Animals were housed separately throughout the study to allow for accurate daily monitoring of food and water intake and stool output. Animals used in this study had no diarrhea or other digestive infections and had not been administered with any antibiotics or other drugs for at least 2 months prior to digests’ collection.

Briefly, three different fresh feces from the same animal were collected, pooled, and homogenized in sterile potassium phosphate buffer (0.1 M, pH 7.2) containing 15% glycerol (*v*/*v*). The homogenized samples were then immediately dispensed into cryotubes. Next, feces were sampled from five animals of each species, respectively. A total of 15 samples were sampled for high-throughput sequencing. Samples were placed in liquid nitrogen immediately after collection, and then transferred to −80 °C ultra-low-temperature freezer for storage. All the samples were subjected to DNA extraction within three days after collection.

### 2.2. DNA Extraction and High-Throughput Sequencing

Total DNA was extracted from all GIT samples (approximately 200 mg per sample) based on repeated bead beating using a mini-bead beater (Biospec Products, Bartlesville, OK, USA) [25]. The integrity of the extracted DNA was measured by electrophoresis on 0.8% agarose gels, and the quality and quantity were determined using a Nanodrop ND-1000 (Termo Scientifc, Wilmington, DE, USA). One-step PCR targeting the V1-V2 region of the 16S rRNA gene was performed with the method of Allali et al. (2015) [26], while ITS genes were amplified using fungal primers ITS1F-ITS1R [27]. All PCR reactions were carried out with 15 μL of Phusion^®^ High-Fidelity PCR Master Mix (New England Biolabs, Ipswich, MA, USA), 2 μM of forward and reverse primers, and about 10 ng template DNA. Thermal cycling consisted of initial denaturation at 98 °C for 1 min, followed by 30 cycles of denaturation at 98 °C for 10 s, annealing at 50 °C for 30 s, and elongation at 72 °C for 30 s. Finally, 72 °C for 5 min.

The same volume of 1X loading buffer (contained SYB green) was mixed with PCR products and operated on via electrophoresis on 2% agarose gel for detection. PCR products were mixed in equidensity ratios. Then, the mixed PCR products were purified with Qiagen Gel Extraction Kit (Qiagen, Hilden, Germany).

Sequencing libraries were generated using TruSeq^®^ DNA PCR-Free Sample Preparation Kit (Illumina, San Diego, CA, USA) by following the manufacturer’s recommendations and index codes were added. The library quality was assessed on the Qubit@ 2.0 Fluorometer (Thermo Scientific, Waltham, MA, USA) and Agilent Bioanalyzer 2100 system. Finally, the library was sequenced on an Illumina NovaSeq platform and 250 bp paired-end reads were generated.

### 2.3. Reads Assembly and Quality Control

We obtained sequencing reads from the fecal sample. Firstly, we assigned paired-end reads to samples based on their unique barcode and truncated by cutting off the barcode and primer sequence. Secondly, the paired-end reads were merged using FLASH [28] (V1.2.7, http://ccb.jhu.edu/software/FLASH/ accessed on 8 October 2020), and the splicing sequences were called raw tags. Thirdly, quality filtering on the raw tags was performed under specific filtering conditions to obtain the high-quality clean tags according to the QIIME2 [29] quality-controlled process. Fourthly, the tags were compared with the reference database (Silva database), using UCHIME algorithm to detect chimera sequences. After removing the chimera sequences, the effective tags were finally obtained.

### 2.4. OTU Cluster, Species Annotation, and Function Prediction

Sequence analyses were performed by Uparse software [30] (Uparse v7.0.1001). Sequences with ≥97% similarity were assigned to the same OTUs. A representative sequence for each OTU was screened for further annotation. For each representative sequence, the Silva database [31] was used based on the Mothur algorithm and the Unit (v8.2) database (https://unite.ut.ee/ accessed on 15 October 2020) was used by the blast method of Qiime2 to annotate bacterial and fungal taxonomic information, respectively. OTU abundance information was normalized using a standard of sequence numbers corresponding to the sample with the fewest sequences. Subsequent analysis of alpha diversity and beta diversity were all performed based on these output normalized data. The bacteriome and mycobiome functional profiles were predicted based on Tax4Fun and FUNGuild2, respectively. 

### 2.5. Alpha Diversity and Beta Diversity

Alpha diversity was applied in analyzing the complexity of species diversity for a sample and the indices like ACE were calculated with QIIME2 and displayed with R software (v4.0.3).

Beta diversity analysis was used to evaluate the differences in the samples in species complexity. Beta diversity on both weighted and unweighted unifrac was calculated by QIIME2 software (https://qiime2.org/accessed on 15 October 2020). Unweighted pair-group method with arithmetic means (UPGMA) clustering was performed as a type of hierarchical clustering method to interpret the distance matrix using average linkage and was conducted by QIIME2 software.

### 2.6. Statistical Analysis and Graphing

Through the above calculating, we could obtain several tables including the microbial composition and function annotation distribution in each sample. Using these data, we compared the commonality and peculiarity between different samples and feeding diet with multiple methods. General manipulation and basic analyses of the dataset were performed in Origin (v2022), R, and Cytoscape (v3.8.1) softwares.

## 3. Results

### 3.1. Taxonomic Assignment of the Gut Mycobiome and Bacteriome among Three Special Laboratory Animals

After collecting the fecal samples from marmots (*n* = 5), chinchillas (*n* = 5), and ferrets (*n* = 5), we analyzed the composition of the gut bacteriome and mycobiome using 16S rRNA and ITS sequencing. The digestive tract structures of the three species of animals are shown in Figure 1. We obtained a total of 1,261,965 16S rRNA reads and 1,295,385 ITS reads, with an average sequencing depth of 84,131 reads/sample for the bacteriome and 86,359 reads/sample for the mycobiome. The sequencing quality and quantity were sufficient for this study (Appendix A). These sequence data were annotated to operational taxonomic units (OTUs) using the Silva and Unit databases, and the number of reads for each sample is provided in the Appendix A.

We identified a total of 3193 bacterial OTUs and 924 fungal OTUs. Compared to the fungi, the bacteria not only had more annotations, but also had a larger proportion of OTUs shared by the three animal species (>50%) (Figure 2a and Figure 3a). Among the three animal species, the chinchilla had a higher number of bacterial OTUs compared to the other two species, which was consistent with the subsequent bacteria annotation results. The annotation numbers for microorganisms at each level were 37 (phylum), 61 (class), 135 (order), 240 (family), 481 (genus), and 312 (species) for the bacteria, and 10, 30, 75, 181, 321, and 436 for the fungi, respectively. The distribution of bacterial OTUs at the genus level is shown in Figure 2b and Figure 3b, and additional taxonomic levels are shown in Appendix A. The Shannon and ACE indices were used to represent the diversity and richness of the microbial communities, respectively. The ferret had lower alpha diversity for both the bacteria and fungi compared to the other two animals (Figure 2c and Figure 3c).

Based on the annotation results of all the samples and the abundance information of the OTUs, we combined the OTU information of the same classification to obtain the species-richness information table. In the analysis of the microbial composition, we analyzed the composition of microorganisms at each taxonomic level. At the phylum level, the dominant bacteria were Firmicutes (46.38%), Bacteroidetes (24.20%), and Proteobacteria (20.72%) (Figure 2d). The most abundant genus was Clostridium (Figure 2e). More results at other taxonomic levels are shown in Appendix A. Ascomycota was the dominant phylum in the fungi, with a proportion exceeding 80% (Figure 3d). The composition of microorganisms at the genus level is shown in Figure 3e. Fungal microorganisms showed significant differences at the genus level. Among the ten most abundant genera, *Acaulium*, *Piromyces*, and *Kernia* were more abundant in the marmot, while *Teunomyces*, *Penicillium*, and *Gamsia* were more abundant in the chinchilla. *Kurtzmaniella*, *Debaryomyces*, *Rhodotorula*, and *Eleutherascus* were mainly distributed in the ferret. The results at other taxonomic levels are shown in Appendix A. Meanwhile, the top 20 genera with an average abundance ranking were focused on three phyla for both the bacteria (Proteobacteria, Firmicutes, Bacteroidota) and fungi (Neocallimastigomycota, Ascomycota, Basidiomycota) (Figure 2f and Figure 3f). The ferret had a higher abundance of *Clostridium* belonging to *Firmicutes*. Compared to the bacteria, the fungal microorganisms showed significant differences. *Piromyces*, an anaerobic fungus under Neocallimastigomycota, only existed in the marmot. *Rhodotorula* and *Wardomyces*, both under Basidiomycota, were mainly distributed and non-existent in the ferret, respectively. Other genera were differently distributed among the different animal species and were all part of Ascomycota.

### 3.2. Structural Comparison of Gut Mycobiome and Bacteriome from Different Digestive Tract Structures in Laboratory Animals

To further analyze the microbial community diversity among the different animal species, the unweighted unifrac distance was calculated based on the phylogenetic relationship between the OTUs. Then, using the abundance information of the OTUs, the weighted unifrac distance was constructed. The unifrac distance showed that the chinchilla and marmot had smaller differences (Figure 4a,d). The results of the unweighted pair-group method with arithmetic mean (UPGMA) disclosed the predominant bacterial and fungal phyla (Figure 4b,e). The cluster trees implied that the chinchilla and marmot had a closer relationship, which was consistent with them both belonging to the rodentia order, while the ferret belonged to carnivora. The top 10 species with the highest average abundance ranking were selected from the three groups of samples at each classification level, and the ternary plots were generated to reveal the absolute differences in dominant species among the three groups at different taxonomic levels (Figure 4c,f). The dominant microbes at different levels and their distribution among the three animal species are shown in Appendix A.

To further explore the differences in microorganisms between the animal species and their feeding habits, we performed LEfSe analysis on the bacteria and fungi separately (Figure 5). To identify the key differential microorganisms, we also considered abundance information, including higher abundance and consistent distribution. Firstly, we identified the important bacteria for the three animal species (Figure 5a and Appendix A). In the chinchilla, Bacteroidia and Bacteroidales were significantly abundant bacteria at the class and order level. The genera *Prevotella*, *Desulfovibrio*, and *Parabacteroides* were the more abundant bacteria and were also under the order Bacteroidales. Species such as *Akkermansia muciniphila*, *Parabacteroides chinchillae*, *Alistipes inops*, and *Pseudoxanthomonas mexicana* had significantly higher abundance than the other animal species at the species level. In the marmot, the genus *Methanobrevibacter*, including its higher taxonomic levels such as family Methanobacteriaceae, order Methanobacteriales, class Methanobacteria, and phylum Euryarchaeota, were all significantly abundant. Additionally, the families Muribaculaceae and Paludibacteraceae were significantly abundant bacteria. Species such as *Pseudomonas pertucinogena* and *Parabacteroides goldsteinii* were also more abundant. In the ferret, there were more abundant bacteria. The order Clostridiales was the most significantly abundant bacteria (LDA = 8.24), mainly influenced by the species *Clostridium perfringens* and *Clostridium colicanis* under the *Clostridium* genus. Similar conditions occurred for species such as *Vagococcus teuberi* under *Vagococcus* and *Enterococcus faecium* under *Enterococcus*, resulting in differences at the family level for Vagococcaceae and Enterococcaceae, respectively. Additionally, *Pseudomonas litoralis* and *Romboutsia ilealis* under *Pseudomonas* and *Romboutsia*, respectively, were all significantly abundant. The abundance of species such as *Chryseobacterium balustinum* under the order Flavobacteriales and the genus *Jeotgalibaca* were also significantly higher in the ferret. All the bacteria at different levels mentioned above were significantly abundant in the corresponding animal species, as reflected in the following LEfSe analysis results.

Compared with the bacteria, the fungi had fewer microorganisms with differences in abundance (Figure 5b and Appendix A). In the chinchilla, *Teunomyces aglyptinius* and *Ascochyta medicaginicola* under the genera *Teunomyces* and *Ascochyta*, respectively, were both significantly abundant. *Microascus brevicaulis* was more abundant in the marmot. *Kernia* and *Candida* were two other fungal microorganisms at the genus level. Moreover, at the kingdom level, the fungi were also more abundant, indicating the presence of more unknown fungal microorganisms in the marmot. As for the ferret, *Kurtzmaniella* and *Cephaliophora tropica* were the more abundant fungi at the genus and species level, respectively.

To explore the correlation between feeding habits and microbial communities, we grouped the chinchilla and marmot as herbivores, while the ferret was classified as a carnivore (Figure 5c and Appendix A). *Odoribacter* under Marinifilaceae, a genus of *Bacteroidia* in the order Bacteroidales, was significantly abundant in the herbivores. *Parabacteroides* sp. CT06, *Parabacteroides goldsteinii*, and *Parabacteroides chinchillae* under the family Tannerellaceae were three more abundant bacterial species. In addition to these bacterial microorganisms, *Desulfovibrio*, a genus of Desulfovibrionia in the family Desulfovibrionaceae, *Prevotella* under Muribaculaceae, *Alistipes inops* under Rikenellaceae, Akkermansiaceae under Verrucomicrobiales, Flavobacteriaceae, and *Lactobacillus* were all more abundant bacteria at different levels. In the carnivores, *Clostridium colicanis*, *Clostridium perfringens*, and *Clostridium baratii* were three species of *Clostridium* in the order Clostridiales, which was the most significantly abundant bacteria (LDA > 8). *Chryseobacterium balustinum* under Flavobacteriales was another significant species. *Enterococcus faecium* and *Vagococcus teuberi*, which were more abundant, further promoted the differentiation at both the genus and family levels. Other species such as *Paeniclostridium sordellii*, *Romboutsia ilealis*, *Acinetobacter lwoffii*, and *Lactococcus garvieae* were key bacteria that caused differences at higher levels (genus and family for the first two and last two, respectively). Apart from the previously mentioned interactive bacteria, Nocardiaceae, Mycoplasmataceae, *Kurthia*, and *Jeotgalibaca* were other more abundant families or genera in the carnivores.

The fungal microorganisms showed less variation in their feeding habits compared to the bacteria, as shown in Figure 5d and Appendix A. Species such as *Microascus brevicaulis* and *Candida* sp. were more abundant in herbivores. These species were also more diverse in the marmots, which contained a higher number of unknown fungal microorganisms. In carnivores, *Cutaneotrichosporon curvatus*, a species of Trichosporonales in the family Trichosporonaceae, was significantly more abundant.

### 3.3. Correlation Analysis of the Shared and Distinct Mycobiome and Bacteriome Compositions in Three Laboratory Animal Species

Due to the distinct co-occurrence relationships among microorganisms in different environments, the species co-occurrence network can visually show the impact of different environmental factors on microbial adaptability, as well as the dominant species and closely interacting species groups in a particular environment. We selected the top 100 microbial genera and calculated the Spearman correlation coefficients. We considered connections with coefficients greater than 0.6 or less than −0.6 and *p*-values less than 0.05.

Firstly, we obtained the co-occurrence network within the bacteria (Figure 6a) and fungi (Figure 6b) separately. There were more connections between the bacteria compared to the fungi, and most of the genera belonged to Proteobacteria, Firmicutes, and Bacteroidota. *Anaeroplasma*, Lachnospiraceae incertae sedis, *Jeotgalicoccus*, *Marinospirillum*, *Acholeplasma*, Oscillospiraceae incertae sedis, *Ruminococcus*, and *Fibrobacter* had more edges in the network, indicating their important role in maintaining the stability of the microbial community structure and function. In comparison, the fungi had fewer connections between their genera, mainly belonging to Ascomycota and Basidiomycota. *Chaetomium*, *Fusarium*, *Teunomyces*, *Ascochyta*, *Enterocarpus*, *Cephalotrichum*, *Botryotrichum*, *Myrothecium*, *Vishniacozyma*, and *Omnidemptus* were the core genera nodes in the co-occurrence network. Overall, the bacteria exhibited a more complex network pattern compared to the fungi, with 96 out of 100 genera having connections with others (compared to the fungi with 46 out of 100). Most associations were positively correlated for both the bacteria (601 out of 697) and fungi (97 out of 106).

To explore the connections between the bacteria and fungi, we calculated the correlation coefficient and constructed networks for the three animal species (Figure 7a–c). The majority of genera belonged to Proteobacteria, Firmicutes, Bacteroidota, Ascomycota, and Basidiomycota. In the chinchilla, the network had a relatively rich number of connections, consisting of 92 bacteria and 65 fungi. The important genera with more degrees in the graph were *Microascus*, *Romboutsia*, *Mucispirillum*, *Teunomyces*, *Fusicolla*, *Dactylonectria*, *Lachnoanaerobaculum*, *Proteiniphilum*, *Kurtzmaniella*, *Filobasidium*, *Omnidemptus*, *Marinospirillum*, and *Anthodidymella*. Similarly, in the marmot network, there were 79 bacteria and 55 fungi. The core genera nodes in the co-occurrence network were *Lactobacillus*, *Massilia*, *Solicoccozyma*, *Aliidiomarina*, *Halomonas*, *Aestuariicella*, *Chaetomium*, *Cutibacterium*, *Lindtneria*, and *Olpidium*. However, the ferret’s network had less than half the number of connections compared to the other two animal species due to a lower number of genera (bacteria:79, fungi:30). In comparison to the marmot and chinchilla, the core genera for the ferret were *Solicoccozyma*, *Mortierella*, *Pseudocoleophoma*, *Psathyrella*, *Plectosphaerella*, *Aspergillus*, *Fusarium*, and *Ascochyta*, which were all fungi. Interestingly, we observed an increasing trend in the number of fungi from the chinchilla to the marmot and ferret. Additionally, the core microorganisms differed between the three animal species. While Firmicutes was the most frequently observed phylum in the core genera of the intra-bacteria network, Proteobacteria played a more important role in the marmot and ferret, specifically *Massilia* and *Comamonas*. Furthermore, the fungi occupied more than half of the core microorganisms, especially in the ferret, where the top eight microorganisms were all fungi.

### 3.4. Functional Genes Analysis Revealed That the Gut Mycobiome–Bacteriome Interface Is Influenced by Environmental Factors and the Structure of the Digestive Tract in Laboratory Animals

To explore the potential functions of the microbial community, we predicted the bacterial and fungal functions using Tax4Fun and FunGuild, respectively. Overall, most genes were annotated to the metabolism. Specifically, membrane transport, translation, replication and repair, carbohydrate metabolism, and amino acid metabolism had higher gene relative abundances (Figure 8a). We also constructed the phylogenetic relationship of the top 30 genera in the bacteria, which highlighted how the differences in major microbial communities contributed to various functional guilds (Figure 8b). Saprotroph was the most abundant function for all three animal species (Figure 8c). Symbiotroph and Pathotroph–Saprotroph, two other abundant functions, were more significant in the marmot and ferret. Additionally, the marmot had many unassigned functions, indicating the presence of many fungi to be studied. Furthermore, we analyzed the phylogeny of the top 20 genera in the fungi, supporting our hypothesis that diet and host phylogeny impose environmental filtering on specific functional guilds and/or certain taxa (Figure 8d). To further identify differences between groups at different classification levels, *t*-tests were conducted to determine functions with significant differences (*p*-value < 0.05). The results showed that the differences increased gradually from chinchilla–marmot, marmot–ferret, to chinchilla–ferret (Appendix A).

## 4. Discussion

This study involved fecal samples from three special animal species (chinchilla, marmot, and ferret) and conducted a comprehensive analysis of the bacteriome and mycobiome. High-throughput sequencing analysis was used to analyze and compare the microbial structure, composition, and function between the animal species, as well as their digestive tract structure and feeding habitat. Moreover, potential associations among the microbial community revealed that the different gut microbiota collectively affect and modulate the physiological and pathological states of the hosts.

In general, the most dominant phyla in the gut microbiota of the three animal species were Firmicutes, Bacteroidota, and Proteobacteria, which were also dominant in mammals [32]. Regarding the fungal phyla, the most abundant fungi in the samples were Ascomycota (>80%), followed by Basidiomycota and Chytridiomycota. There were certainly influences on the microbiota in all three animal species, which were significantly demonstrated for both the bacteria and fungi, regardless of whether they were dominant or differential microbes. In the chinchillas, *Bacteroidales* were more abundant and related to the adaptation of the gut microbiome to a lignocellulose diet [33] and diseases like inflammatory bowel disease [34]. *Prevotella*, a representative genus of Bacteroidetes, was also a key microbe that has been shown to be useful in improving disease prognosis [1,35]. An increased abundance of *Prevotella* was correlated with various amino acid and carbohydrate metabolism pathways, including short-chain fatty acid production [36] and branched-chain amino acid metabolism [37]. *Prevotella* and *Bacteroidales* were known to produce significant amounts of hydrogen in ruminants [38]. *Akkermansia muciniphila*, *Bacteroides*, and *Desulfovibrio* were all core microbes connected with gut microbiota-related metabolites, including short-chain fatty acids, branched-chain amino acids, and aromatic amino acids [39]. Additionally, sulfate-reducing *Desulfovibrio* and methane-producing *Methanobrevibacter* were related to the pectin-enriched diet [40]. *Parabacteroides* was also a diet-derived cholesterol-interacting, urease-related, and short-chain fatty acid-producing bacteria [41]. It has been associated with both beneficial and pathogenic effects on human health [42] and may participate in food preference alterations during obesity, likely through the gut-to-brain axis [43]. Accumulating evidence points to *Akkermansia muciniphila* as a novel candidate for preventing or treating obesity-related metabolic disorders [44]. *Parabacteroides chinchillae* was isolated from chinchilla (*Chincilla lanigera*) feces [45] but has few related studies. *Alistipes inops* under the genus *Alistipes* have emerging implications for inflammation, cancer, and mental health [46]. *Pseudoxanthomonas mexicana*, as the recent origin of the blaAIM-1 carbapenemase, a metallo-β-lactamase that catalyzes the hydrolysis of various β-enzymes of lactam antibiotics, was connected with the immune system of the chinchilla. The genus *Methanobrevibacter* in the marmot was significantly abundant and caused differential effects on its every higher taxonomic level, including the phylum Euryarchaeota. *Methanobrevibacter*, as one of the dominant gut lineage archaeomes in mammals, is driven by several factors such as host phylogeny, diet type, fiber content, and intestinal tract physiology [47].

Furthermore, *Methanobrevibacter* was positively linked to pigs’ body weight and played essential roles in swine nutrition, metabolism, and growth performance [48]. Moreover, although we attribute the marmot to being herbivorous (which will be discussed later), they sometimes eat small animals such as insects and exhibit omnivorous behavior under breeding conditions. A previous study found that another omnivorous animal, lizard individuals, also harbored significantly more *Methanobrevibacter* [49]. Muribaculaceae, a short-chain fatty acid (SCFA)-producing bacteria [50], was associated with a high-fat diet and has been demonstrated in piglets [51]. Paludibacteraceae was another abundant family in the marmot, and it was connected with low protein and branched-chain amino acids in pigs [52]. *Pseudomonas pertucinogena*, a bacterium with biotechnological potential, displayed adaptability to the environment. The gut commensal *Parabacteroides goldsteinii* played a predominant role in the anti-obesity effects of polysaccharides [53]. Clostridiales, under Firmicutes, were the most abundant and differential microbes in the ferret (LDA = 8.24), and both *Clostridium perfringens* and *Clostridium colicanis* contributed to this condition. *Clostridium perfringens*, commonly found in soil, water, and intestines, can produce various toxins and cause necrotic enteritis [54]. *Clostridium perfringens* was the most abundant species in the ferret and may be associated with its feeding habitat. Similarly, *Clostridium colicanis*, which has been connected to the difference between vegetarians and non-vegetarians in humans [55], may also be associated with the ferret’s feeding habitat. *Chryseobacterium balustinum* is a spoilage organism that exists in raw meat. *Enterococcus faecium*, an opportunistic pathogen as well as a probiotic [56], together with the diet, has the function of degrading antinutritional factors and enhancing nutritional value [57]. *Vagococcus teuberi* may be related to spoilage and foodborne diseases [58]. *Pseudomonas litoralis* and *Romboutsia ilealis* are potentially disease-related bacteria. *Jeotgalibaca*, a hydrolytic bacterium, enhances the degradation of sheep manure and increases biogas production [59]. Compared to the bacteria, the abundance and diversity of the fungal community were relatively lower. In chinchillas, *Ascochyta medicaginicola* causes spring black stem and leaf spot of alfalfa and the model legume Medicago truncatula [60], which is related to their feeding habitat. *Kurtzmaniella* has also been studied, and its abundance is affected by diet [61]. As for the marmot, *Microascus brevicaulis* was detected at all composting stages and showed the highest relative abundance [62]. The genus *Kernia* comprises species that are commonly isolated from dung, soil, decaying meat, and animal skin and is associated with biodegradation [63]. *Candida* is another important genus that is connected with abnormal blood glucose levels [64] and rheumatoid arthritis [65]. It is determined by habitual diet and the host’s physiological state [66].

The host species and environmental factors, such as diet, are important in determining the diversity of gut microbiota [67]. In this study, we included chinchillas and marmots as herbivores and ferrets as carnivores. There were differences between the two groups, especially in terms of the bacteria. *Odoribacter*, known for producing short-chain fatty acids [68], has the potential to regulate physiological functions [69]. The Tannerellaceae family was also abundant in herbivores and contained three significantly abundant species. *Parabacteroides goldsteinii* and *Parabacteroides chinchillae* were mostly found in the marmots and chinchillas, respectively, as previously discussed. Another species, *Parabacteroides* sp. CT06, which belongs to the Tannerellaceae family, is linked to the α-ketoacid diet [70].

Akkermansiaceae, which are potentially beneficial bacteria, showed a correlation between fecal fatty acids and glycerolipids, serum glycerophospholipids, and cortical fatty acids [71]. The Flavobacteriaceae family is an important family within the Bacteroidetes phylum. Members of this family are found in a wide variety of freshwater and soil habitats and are also associated with animals and plants. Herbivores were found to have greater exposure to these bacteria, as confirmed by the analysis results. *Lactobacillus* is a key microbe in monogastric animals [72], especially found in the stomach, and plays an essential role in overall host health [73].

Compared to the bacteria, the fungi did not exhibit significant differences in abundance. However, certain traits of the fungi, like the superficial and invasive infections of *Microascus brevicaulis* [74], such as the ability of *Candida* species to generate biofilms, can contribute to abundance and pathogenicity [75]. In carnivores, *Clostridium baratii*, an anaerobic bacterial species, was one of the three abundant species of *Clostridium* found in higher quantities than in herbivores. Through the fermentation of glucose in peptone yeast glucose broth, it can produce lactic acid, acetic acid, butyric acid, and possibly hydrogen gas. Similar to the other two species of *Clostridium* mentioned, *Clostridium baratii* aids in carnivorous digestion and also produces toxins that can cause botulism [76].

The genus *Kurthia*, similar to the *Chryseobacterium balustinum* mentioned earlier, potentially has the ability to degrade cholesterol and contribute to spoilage [77], while the Nocardiaceae family is associated with steroid hormone synthesis [78]. *Cutaneotrichosporon curvatus*, an oleaginous yeast, has the ability to produce substantial amounts of polyunsaturated fatty acids [79] and likely plays an important role in the host’s metabolism and immune system. Other bacteria such as *Paeniclostridium sordellii*, *Acinetobacter lwoffii*, *Lactococcus garvieae*, and the Mycoplasmataceae family are potentially associated with diseases.

Generally, many pathogenic bacteria within a healthy gut do not cause problems in immune-competent hosts. One important reason for this is the antagonistic and synergistic effects between microorganisms. Therefore, we conducted a study to explore the network dynamics of intra- and inter-dominant bacterial and fungal genera. Overall, we observed the network topology formed by the expansion of taxa within Firmicutes or Ascomycota, indicating their leading roles in the cross-talk of the bacterial and fungal community. The close connections among microbes, especially bacteria, demonstrated potential surprising connections between different biological functions. The top 10 genera in the intra-bacteria network were associated with several functions, including disease, environmental adaptation, diet, and immune responses. The results showed that the core nodes of the bacteria-associated network were harmful microorganisms, and the fungi played a similar role. We propose that harmful microbes have more opportunities to enter the gut due to frequent contacts with the wild, and more beneficial microbes are needed to antagonize them and maintain the stability of the host’s metabolism and immune system. The increasing number of fungi from chinchilla to marmot to ferret indicated a connection with their feeding habitat. In the ferrets, the degree of Solicoccozyma under Basidiomycota in the bacteria–fungi network was almost two times higher than the second genus, and some studies suggest that it may be associated with the degradation of dietary carbohydrates.

Our functional analysis indicated that the gut microbiota in all three animal species exhibited high metabolic activity, especially in carbohydrate and amino acid metabolism. We also found that the differences between the chinchilla and ferret were greater than the differences between the other two pairs, which is consistent with their dietary differences. The analysis of microbial differences between groups at different classification levels further confirmed these differences. There were many unassigned functions for the marmots, indicating the presence of a large number of unknown taxa. These taxa may play an important role in adapting to digestion and warrant further exploration. Furthermore, the mycobiome showed greater differences among individuals compared to the bacteriome, likely due to its lower proportion in the gut microbiota (<1%) and its susceptibility to the external environment. Therefore, this phenomenon inspires us to consider multiple factors in the study of the mycobiome, whether in breeding or scientific research.

Our findings make a significant contribution to understanding the impact of different microbial communities on adapting to different physiological structures and dietary habits. These results not only provide a basis for further research on the regulatory effects of the gut microbiota on the host but also provide basic data for comparative medical research. Furthermore, we provide high-throughput microbiome data that can be used as a springboard to uncover the potential of the microbiome in animal model research against infections.

## 5. Conclusions

As we combined different public datasets to identify relationships between animals and the microbiome, the comparison between studies is likely constrained by technical variation. In addition, the included studies applied 16S and ITS amplicon sequencing strategies, which do not reliably provide species- and strain-level taxonomic resolution. To weaken some of these limitations, we applied the Tax4Fun2 with a wider reference dataset and more accurate prediction methods. In this paper, we comprehensively described the structure of the microbiota and their network relationships across three different animal species—chinchilla, marmot, and ferret—using high-throughput sequencing. Our findings revealed that the structure of the gut microbiota in these animals is influenced by their digestive tract structure and feeding habitat. Despite the fungi being relatively less abundant, they still play significant roles in modulating the host’s metabolism. Bacteria, along with fungi, maintain a balance between the environment, diet, and host through complex antagonistic and synergistic effects among microorganisms. Furthermore, the composition of intestinal flora also affects the selection of appropriate animal models for different pathogen infections. In conclusion, these results provide a basis for further medical research on the impact of the host microbiome on infection outcomes in animal models.

## Figures and Tables

**Figure 1 microorganisms-12-00646-f001:**
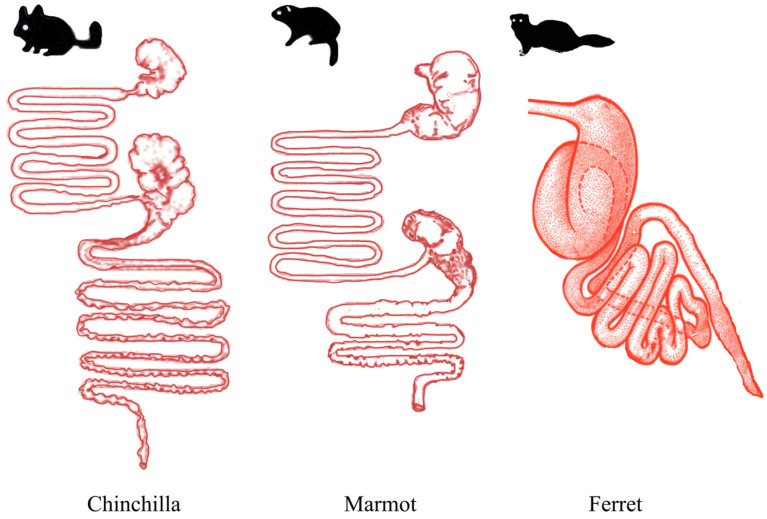
Graphical digestive tract structure representation of chinchilla, marmot, and ferret. Reference website https://www.cnsweb.org/digestive_system_of_vertebrates/ (accessed on 15 September 2023) drawing the figure plate.

**Figure 2 microorganisms-12-00646-f002:**
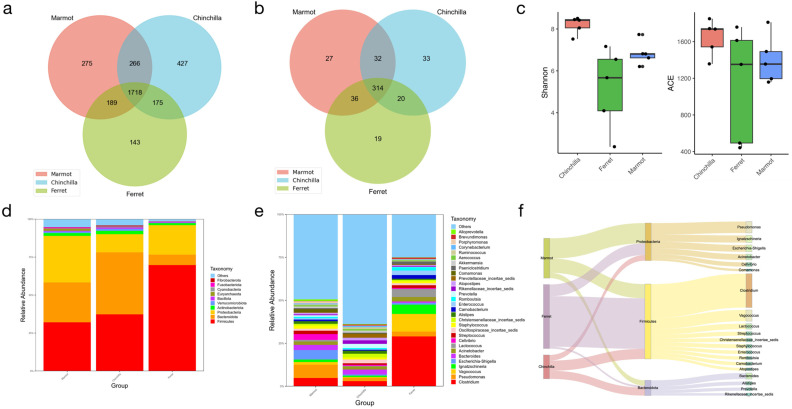
Comparison of the three animal species of bacteria. (**a**) The common and unique OTUs among different groups. (**b**) Comparison of alpha diversity indices. (**c**) The genus composition shared by three animal species. (**d**,**e**) The bacterial compositions at the phylum and genus level. (**f**) Sankey analysis of bacteria.

**Figure 3 microorganisms-12-00646-f003:**
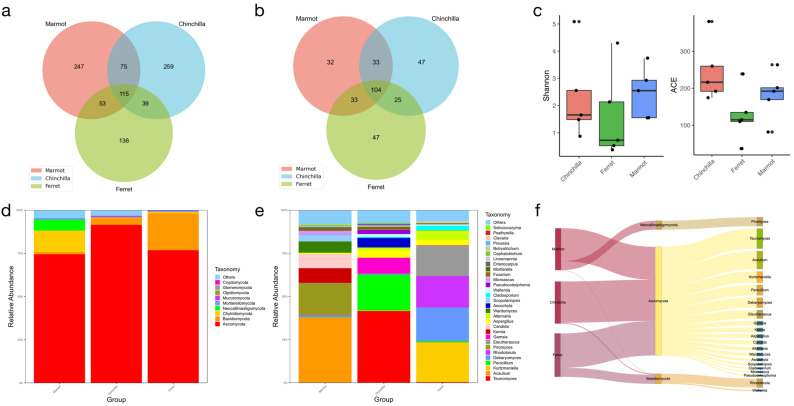
Comparison of the three animal species of fungi. (**a**) The common and unique OTUs among different groups. (**b**) Comparison of alpha diversity indices. (**c**) The genus composition shared by three animals. (**d**,**e**) The fungal compositions at the phylum and genus level. (**f**) Sankey analysis of fungi.

**Figure 4 microorganisms-12-00646-f004:**
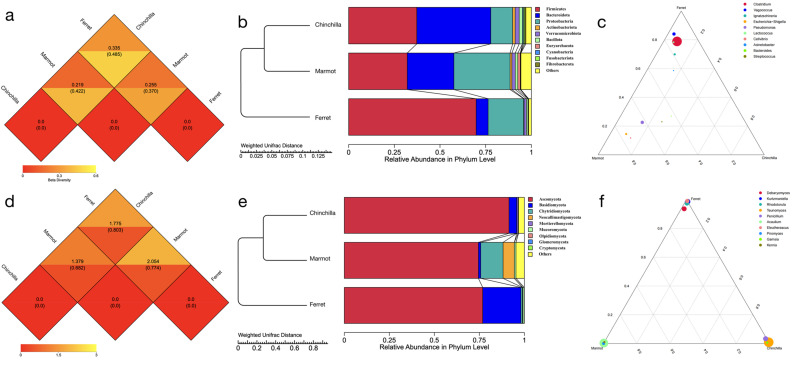
Composition differences of bacteria (**top**) and fungi (**bottom**) for three animal species. (**a**,**d**) Heatmap showing significantly different animal species. The smaller dissimilarity coefficient means the smaller the difference between animal species; in the same grid, the upper and lower values represent the weighted unifrac and unweighted unifrac distances, respectively. (**b**,**e**) Unweighted pair-group method with arithmetic mean (UPGMA) analysis. Microbiota profile clustering based on Bray–Curtis dissimilarities was set to form an OTU hierarchical clustering tree of the overall grouped distribution of microbial species in the gut of each host species. The legend boxes list the 10 most abundant taxa at the phylum level. (**c**,**f**) The ternary plot at the genus level.

**Figure 5 microorganisms-12-00646-f005:**
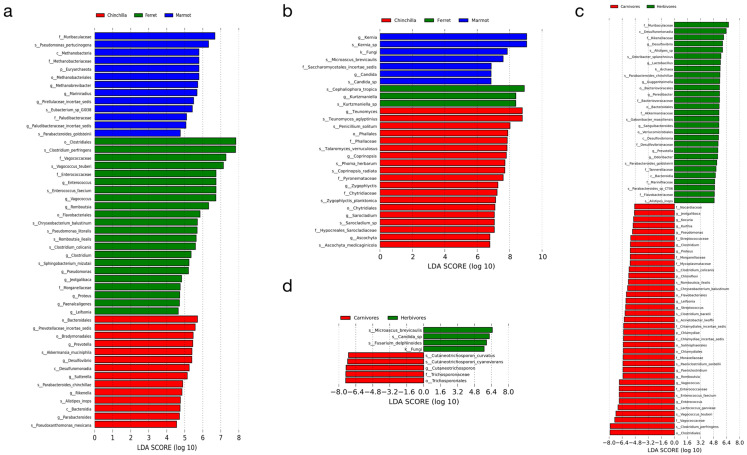
Linear discriminant analysis (LDA) effect size (LEfSe) analysis. Comparison of differential microbiota in three animal species of bacteria (**a**) and fungi (**b**). Comparison of differential microbiota in different feeding habits for bacteria (**c**) and fungi (**d**).

**Figure 6 microorganisms-12-00646-f006:**
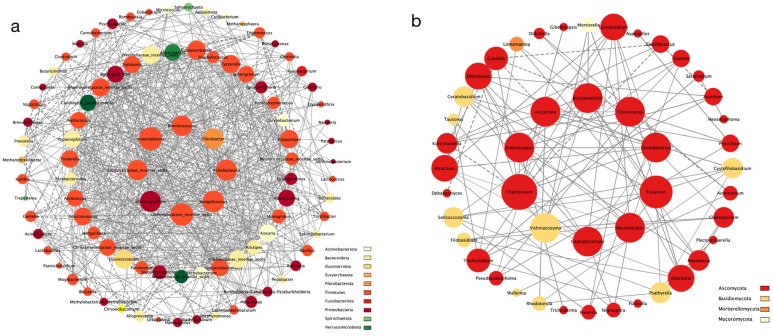
The network analysis within bacteria (**a**) and fungi (**b**). Different node color denotes varied phyla taxa. The weighted node size was based on the connections with other nodes. The weighted edges indicate the correlation coefficient and the thickness of the edge represents the size of the coefficient. Solid lines indicate positive correlation, dashed lines indicate negative correlation.

**Figure 7 microorganisms-12-00646-f007:**
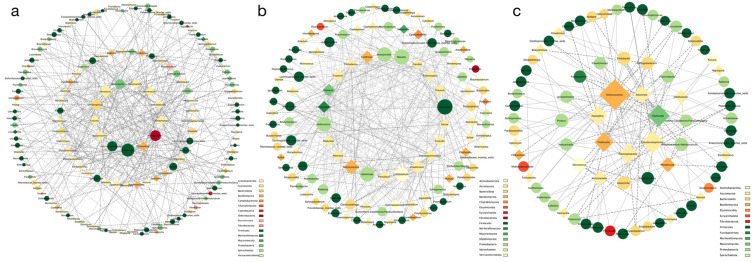
The network analysis between bacteria and fungi in the marmot (**a**), chinchilla (**b**), and ferret (**c**). Ellipse indicates bacteria and diamond indicates fungi. Different node color denotes varied phyla taxa. The weighted node size was based on the connections with other nodes. The weighted edges indicate the correlation coefficient and the thickness of the edge represents the size of the coefficient. Solid lines indicate positive correlation, dashed lines indicate negative correlation.

**Figure 8 microorganisms-12-00646-f008:**
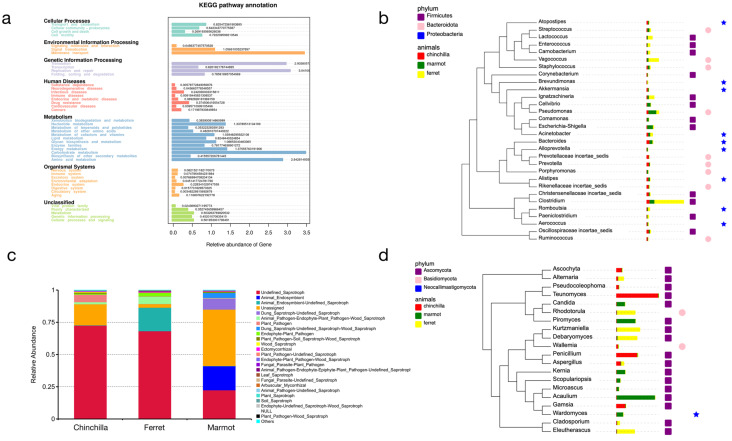
Function annotation and the phylogeny for bacteria (**top**) and fungi (**bottom**). The overview of bacterial function (**a**) and the distribution of fungal functions of three animal species (**c**). The phylogeny is the same as shown in (**b**,**d**). From left to right, the data mapped onto the tree are host- and phylum-level classification.

## Data Availability

The datasets supporting the conclusions of this article are available in the (NMDC10018413) repository, unique persistent identifier, and hyperlink to dataset(s) in https://nmdc.cn/resource/genomics/project/detail/NMDC10018413 (accessed on 27 April 2023). The datasets supporting the conclusions of this article are included within the article and its additional files.

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
