# Peer review of "Comparative Analysis of Gut Microbiomes in Laboratory Chinchillas, Ferrets, and Marmots: Implications for Pathogen Infection Research"

_microorganisms, 2024, doi:10.3390/microorganisms12040646_

Round 1

Reviewer 1 Report

Comments and Suggestions for Authors

This is a well-structured manuscript with the results presented clearly.

The methodology is well-written and clear.

The  discussion could benefit from a succinct comparison to other studies, e.g., in other canines.

It would be recommended that the authors add 2-3 sentences with the limitations of their study just prior to the conclusions section.

The authors suggest in the Introduction that the findings can lay the ground for future applications in medical research. While this is again mentioned in the discussion, it would be preferable if it were expanded by 2-3 sentences with a potential example as to how this may be the case.

Author Response

Reviewer 1:

Q1. It would be recommended that the authors add 2-3 sentences with the limitations of their study just prior to the conclusions section.

R: We sincerely appreciate for your valuable insightful input. We have added the limitations of their study just prior to the conclusions section. “As we combined different public datasets to identify relationships between animals and microbiome, the comparison between studies is likely constrained by technical variation. In addition, the included studies applied 16S and ITS amplicon sequencing strategies which do not reliably provide species and strain level taxonomic resolution. To weaken some of these limitations, we applied the Tax4Fun2 with wider reference dataset and more accurate prediction methods.”

Q2. The authors suggest in the Introduction that the findings can lay the ground for future applications in medical research. While this is again mentioned in the discussion, it would be preferable if it were expanded by 2-3 sentences with a potential example as to how this may be the case.

R: Thank you for your timely advice. We have expanded this part as “Furthermore, the change of host health status reflects the complex balance relationship between host, symbiotic flora and pathogen. Usually, gut microbes can directly interact with pathogens and directly inhibit or promote the process of virus invasion or replication. At present, the impact of the host microbiome on infection outcomes has not been explored in animal models, partly due to the lack of a comprehensive understanding of microbial communities across different laboratory animal species.” in introduction.

Reviewer 2 Report

Comments and Suggestions for Authors

The study utilizes high-throughput sequencing to analyze the bacteriome and mycobiome in fecal samples from chinchilla, marmot, and ferret. The findings underscore the impact of digestive tract structure and feeding habits on the composition of gut microbiota, emphasizing the crucial roles of bacteria and fungi in maintaining a delicate balance vital for host metabolism. The research offers details for future medical studies concerning the influence of host microbiomes on infection outcomes in animal models.

Minor suggestions for enhancing clarity and precision:

1. Revise the title to "Comparative Analysis of Gut Microbiomes in Laboratory Chinchillas, Ferrets, and Marmots: Implications for Pathogen Infection Research" for specificity about the studied animal species.

2. Correct the spelling of "Penicillum" to "Penicillium" in Line 18 for accuracy.

3. Add "chinchillas, ferrets, marmots" to the keywords in Lines 27-28.

4. Consider rephrasing Lines 32-33 for better flow, such as "vital roles in scientific research, teaching, production, verification, safety evaluation, and achievement assessment."

5. Explain the link between fungi and feeding habitat in Lines 70-71.

6. Clarify your wording to "...three animal species..."

7. Provide references for the laboratory animals used (chinchilla, marmot, and ferret), specifying their scientific names, numbers, and origins in the section starting from Line 85.

8. Revise wording in Lines 85-86 to “Information on fecal samples, including age, sex, and sampling date, is provided in Table S1. ”

9. Mention the specific DNA extraction method used in Lines 98-99.

10. Include the corresponding fungal phyla and their percentages in Lines 187-189 for symmetry.

11. Discuss potential reasons for the higher abundance of Methanobrevibacter in omnivorous animals, referencing the study on lizard individuals in Line 419.

Author Response

Reviewer 2:

The study utilizes high-throughput sequencing to analyze the bacteriome and mycobiome in fecal samples from chinchilla, marmot, and ferret. The findings underscore the impact of digestive tract structure and feeding habits on the composition of gut microbiota, emphasizing the crucial roles of bacteria and fungi in maintaining a delicate balance vital for host metabolism. The research offers details for future medical studies concerning the influence of host microbiomes on infection outcomes in animal models. Minor suggestions for enhancing clarity and precision:

Q1. Revise the title to "Comparative Analysis of Gut Microbiomes in Laboratory Chinchillas, Ferrets, and Marmots: Implications for Pathogen Infection Research" for specificity about the studied animal species.

R: Thank you for your timely advice and revised as suggestion.

Q2. Correct the spelling of "Penicillum" to "Penicillium" in Line 18 for accuracy.

R: Yes, revised as suggestion.

Q3. Add "chinchillas, ferrets, marmots" to the keywords in Lines 27-28.

R: Yes, revised as suggestion.

Q4. Consider rephrasing Lines 32-33 for better flow, such as "vital roles in scientific research, teaching, production, verification, safety evaluation, and achievement assessment."

R: Thank you for your valuable advice. We have rephrased this sentence.

Q5. Explain the link between fungi and feeding habitat in Lines 70-71.

R: Yes, we have revised this sentence as “In fact, fungi, as key microbes closely related to feed habitat, with most foods positively or negatively associated, can affect animal growth, development, systemic evolution, nutrient acquisition, cellulose degradation, and fermentation.”

Q6. Clarify your wording to "...three animal species..."

R: Yes, we have revised in the whole manuscript.

Q7. Provide references for the laboratory animals used (chinchilla, marmot, and ferret), specifying their scientific names, numbers, and origins in the section starting from Line 85.

R: Thank you for your kind reminder. We have added these informations “The fecal samples information, including age, sex, and sampling date, was listed in Table S1 from fifteen healthy animals of three species (Chinchilla lanigera, Marmota bobak and Mustela putorius furo) and were raised in Institute of laboratory animal sciences, Chinese academy of medical sciences”.

Q8. Revise wording in Lines 85-86 to “Information on fecal samples, including age, sex, and sampling date, is provided in Table S1.”

R: Yes, revised as suggestion.

Q9. Mention the specific DNA extraction method used in Lines 98-99.

R: Yes, we have added the DNA extraction method “Total DNA was extracted from all GIT samples (approximately 200 mg per sample) based on repeated bead beating using a mini-bead beater (Biospec Products, Bartlesville, USA) [25]. Te integrity of the extracted DNA was measured by electrophoresis on 0.8% agarose gels, and the quality and quantity were determined using a Nanodrop ND-1000 (Termo Scientifc, Wilmington, USA)”.

Q10. Include the corresponding fungal phyla and their percentages in Lines 187-189 for symmetry.

R: There are fewer intestinal fungal phyla than bacterial. Ascomycota was the dominant phylum in intestinal fungal, with a proportion exceeding 80%. The proportion of other phylum is very small.

Q11. Discuss potential reasons for the higher abundance of Methanobrevibacter in omnivorous animals, referencing the study on lizard individuals in Line 419.

R: Methane bacteria can only use very simple substances such as CO2, H2, formic acid, acetic acid and methylamine as energy sources. These simple substances must be broken down by other fermentative bacteria to provide complex organic matter to the methane bacteria, so the methane bacteria must wait until other bacteria have grown in large numbers before they can grow. This may be related to the higher bacterial diversity in the omnivores' gut.